# Two *Clostridium*
*perfringens* Type E Isolates in France

**DOI:** 10.3390/toxins11030138

**Published:** 2019-03-01

**Authors:** Laure Diancourt, Jean Sautereau, Alexis Criscuolo, Michel R. Popoff

**Affiliations:** 1CNR Bactéries anaérobies et Botulisme, Institut Pasteur, 75015 Paris, France; laure.diancourt@pasteur.fr (L.D.); jean.sautereau@pasteur.fr (J.S.); 2Hub de Bioinformatique et Biostatistique—C3BI, Institut Pasteur, USR 3756, CNRS, 75015 Paris, France; alexis.criscuolo@pasteur.fr; 3Bacterial Toxins, Institut Pasteur, ERL 6002, 75015 Paris, France

**Keywords:** *Clostridium perfringens*, iota toxin, *C.**perfringens* enterotoxin, enterotoxemia, food poisoning

## Abstract

*Clostridium**perfringens* type E is a less frequently isolated *C.*
*perfringens* type and has not previously been reported in France. We have characterized two recent type E isolates, *C.*
*perfringens* 508.17 from the intestinal content of a calf that died of enterotoxemia, and 515.17 from the stool of a 60-year-old woman, subsequent to food poisoning, which contained the plasmid pCPPB-1 with variant iota toxin and *C. perfringens* enterotoxin genes.

## 1. Introduction

*Clostridium perfringens* is a ubiquitous gram-positive, spore forming anaerobic bacterium which produces different toxins and is responsible for various diseases in man and animals such as gangrene, food poisoning, diarrhea, necrotic enteritis, and enterotoxemia. Based on the production of four toxins (alpha, beta, epsilon, and iota), *C. perfringens* was traditionally divided into five toxin types (A to E) [1]. More recently, the *C. perfringens* nomenclature was expanded to seven types (A to G) by adding *C. perfringens* enterotoxin (CPE) and NetB carrying strains as different toxin types [2].

*C. perfringens* type E is characterized by the production of iota toxin. Iota toxin is a binary toxin consisting of a binding component (Ib) and an enzymatic component (Ia) which enters cells and modifies the actin cytoskeleton by ADP-ribosylation of actin monomers [3,4]. *C. perfringens* type E is a less frequently isolated type from clinical samples. This toxin type has been isolated from cases of diarrhea or hemorrhagic enteritis and sudden death in neonatal calves, mainly in the US, and more rarely in other animals such as chickens, lambs, goats, and cows [5,6,7,8,9,10,11,12,13]. In France, *C. perfringens* type E isolation has not been previously reported. Here we describe two recent type E isolates, 508.17 and 515.17.

## 2. Results

### 2.1. Identification of Two C. perfringens Type E and Investigation of Iota Toxin Production

The two *C. perfringens* isolates 508.17 and 515.17 were identified as *C. perfringens* type E by detection of iota toxin genes with routine PCR toxin gene identification of *C. perfringens* [14].

The production of iota toxin in *C. perfringens* 508.17 and 515.17 was investigated by western blotting with specific antibodies against Ia and Ib. The reference type A *C. perfringens* strain ATCC 13124 was used as a control. As shown in Figure 1, both components Ia and Ib were detected in the supernatant of *C. perfringens* 508.17 and 515.17 but not in *C. perfringens* ATCC 13124.

The biological activity was tested on Vero cells. The concentration of culture supernatant yielding 50% rounded cells was 156 ng/mL (total protein) for 508.17 and 650 ng/mL (total protein) for 515.17. The level of cytotoxicity was increased four-fold after α-chymotrypsin treatment of the culture supernatants. In comparison, the 50% cytotoxic activity of purified recombinant iota toxin on Vero cells was obtained with 15 ng/mL Ib and 8 ng/mL Ia (not shown).

Concentrated supernatants of *C. perfringens* 508.17 (5 μg total protein), *C. perfringens* 515.17 (5 μg total protein), and *C. perfringens* ATCC 13124 (5 μg total protein) as control, as well as purified Ia (1 ng) and Ib (1 ng) were run on a 10% SDS-PAGE and transferred on nitrocellulose. Blots were incubated with specific rabbit serum against Ia and Ib, respectively, and then with goat immunoglobulin against rabbit IgG labeled with peroxidase. The bands at 55 kDa in the western blot with anti-Ib and the double bands in the western blot with anti-Ia resulted from partial proteolytic degradation.

### 2.2. In Vitro Actin ADP-Ribosylation

The enzymatic activity of iota toxin in *C. perfringens* 508.17 and 515.17 culture supernatants was tested by in vitro ADP-ribosylation with muscular and cellular actin. As shown in Figure 2, ADP-ribosylation of both muscular and cellular actin was observed with culture supernatants from *C. perfringens* 508.17 and 515.17, as well as the purified Ia control. The type A *C. perfringens* ATCC 13124 did not show any actin ADP-ribosylation.

### 2.3. Whole Genome Sequencing

The two *de novo* assemblies are quite similar at first sight, i.e., 3.47 and 3.58 Mbs, 28.00% and 28.06% GC-content, 3,085 and 3,286 CDS for isolates 508.17 and 515.17, respectively. However, scaffolds from isolate 508.17 contain only one sequence that is almost identical to pCPPB-1 (accession number UWOV01000018; 99.8% similarity), whereas those from isolate 515.17 contain 19 sequences that could be reordered against pCPPB-1 (i.e., UWOU0100067, 108, 087, 023, 106, 043, 047, 005, 018, 003, 036, 090, 086, 068, 013, 006, 014, 007, 101; 99.7% identity), and 7 additional sequences showing strong BLAST similarity (e.g., >99.7% identity) with pJFP838D and pD13118_cpe (i.e., UWOU0100008, 009, 016, 022, 024, 026, 034). This indicates that the strain 515.17 contains two additional plasmids related to pJFP838D and pD13118_cpe, or a plasmid combining these sequences.

Iota toxin Ia and Ib components from 508.17 and 515.17 strains are highly similar (99.73% and 99.83% at the amino acid level), closely similar to those of pCPPB-1 iota toxin components (99.87% similarity), and more distantly related to classical iota toxin (91.2% and 89.7%, respectively) (Table 1). Iota toxin sequences from 508.17 and 515.17 strains show a low relatedness (40–42% identity) to iota toxin variant BEC/CPILE found in contaminated food in Japan [15,16].

The two isolates also contain alpha toxin, perfringolysin, and *cpe* genes. CPE from 508.17 and 515.17 strains are identical at the amino acid sequence level with variant CPE from pCPPB-1, and differ from classical CPE at 12 positions (Table 2).

### 2.4. Phylogenetic Analysis

Phylogenetic analysis of the *C. perfringens* core-genome shows that the two strains 508.17 and 515.17 are related but not identical (Figure 3). Both strains are neighbors within a specific clade, emerging between the type A strain MJR7757A (originated from a human host) and type A JFP strains (originated from foal or dog hosts). However, they are distantly related to the classical *C. perfringens* type E strain JGS1987. Of note, the strain 508.17 is characterized by the absence of the toxin gene *cpb2*, whereas the two type E strains 515.17 and JGS1987 show identical toxin gene content.

## 3. Discussion

Most *C. perfringens* type E strains have been found to contain the iota toxin (*iap* and *ibp*) genes on a plasmid flanked by the insertion sequence IS*1151* and in close proximity to a silent *cpe* gene [17]. Iota toxin plasmids have mostly a pCPF5603 backbone with insertion of *iap* and *ibp* genes within the *cpe* promoter, thus preventing *cpe* transcription in addition to nonsense and frame-shift mutations in the open reading frame (ORF) [13]. More recently, four *C. perfringens* type E strains have been characterized to have *iap* and *ibp* genes on a pCPPB-1-related plasmid. pCPPB-1 is a ~67 kbp plasmid containing 72 putative ORFs that are organized in three regions, a putative replication plasmid and transfer region, a toxin region, and a variable region. pCPPB-1 retains the backbone of pCPF4969 that is the classical plasmid containing *cpe* in *C. perfringens* type A, but lacks IS elements in the vicinity of toxin genes. In contrast to pCPF5603, pCPPB-1 contains a functional *cpe*. Indeed, albeit *iap* and *ibp* genes are inserted into the *cpe* promoter region, only the promoter P3 is missing. The two other *cpe* promoters (P1 and P2) are the major promoters, and are preserved, therefore allowing *cpe* expression. This is in contrast to the strains with iota toxin plasmid of the pCPF5603 family, which produce no CPE even in sporulation conditions [18]. Iota and *cpe* genes harbored on pCPPB-1 plasmid show variations with the corresponding genes on classical strains. Indeed, *iap* and *ibp* from pCPPB-1 share 87% and 89% identity with the corresponding genes of *C. perfringens* E NCIB10748 [18,19], respectively, at the nucleotide level. The variant CPE from pCPPB-1 is 96% identical (10 amino acid differences on 319) to classical CPE [18].

Recently, *C. perfringens* strains isolated from food poisoning outbreaks in Japan were characterized as producing an iota toxin variant called BEC (binary toxin of *C. perfringens*) or CPILE (*Clostridium perfringens* iota-like enterotoxin) [15,16]. These strains lack *cpe* genes and CPE production. The two components BECa/CPILE-a and BECb/CPILE-b share 43% and 41–42% identity with Ia and Ib from the classical *C. perfringens* type E strain NCIB10748, respectively [15,16]. *becA* and *becB* are located on a large size plasmid (pCP-OS1/pCP-TS1, 54,635 bp) for which most parts (69%) are highly similar (92–99% identity) with sequences of the plasmid pCP13 from *C. perfringens* strain 13 [18]. BEC/CPILE is enterotoxic and induces fluid accumulation in rabbit ileal loop and suckling mice [15,18]. BECa/CPILEa ADP-ribosylates all isoforms of actin monomers and retains a similar structure compared to Ia [20]. Iota toxins from strains 508.17 and 515.17 share the same enzymatic activity profile as BEC/CPILE. Indeed, they also ADP-ribosylate both muscular and cellular actin. However, iota toxins from 508.17 and 515.17 as well as the classical iota toxin retain a low level of identity with BEC/CPILE.

The two French *C. perfringens* type E isolates have distinct chromosomal genetic backgrounds related but not identical to those of the variant *C. perfringens* type E [18]. They are distantly related to the classical *C. perfringens* type E and enterotoxigenic and non-enterotoxigenic type A strains, as well as type A strains with chromosomally located *cpe* [21,22] (Figure 3). However, the two French isolates have acquired a pCPPB-1 plasmid similar to that found in the variant *C. perfringens* type E strains isolated from meat products in Japan [18]. The two French isolates 508.17 and 515.17 have distinct origins, from calf and food intoxication in man, respectively. Their relatedness with *C. perfringens* type E strains reported in Japan raises questions as to whether they share a common source, and what the possible mode of dissemination might be.

## 4. Materials and Methods

### 4.1. C. perfringens Isolates

*C. perfringens* 508.17 was isolated from the intestinal content of a 6-month-old calf that died with an enterotoxemia syndrome in central France. The calf received a vaccination against blue tongue one day before. The strain 515.17 was from the stool of a 60-year-old woman in a nursing home showing a *C. perfringens* food poisoning. The stool of two other patients of the same food poisoning outbreak yielded classical enterotoxigenic *C. perfringens* strains with *cpe* gene located on the chromosome.

### 4.2. C. perfringens Cultures

*C. perfringens* strains were grown in trypticase-glucose-yeast extract (20 g of Trypticase, 30 g of yeast extract, and 0.5 g of cysteine hydrochloride per liter, pH 7.2) (TGY) [23] under anaerobic conditions at 37 °C overnight. The culture supernatant was concentrated (about 20-fold) by ammonium sulfate precipitation (70% saturation), centrifuged at 10,000 rpm for 10 min, and then dialysis against Tris-HCl 10 mM, pH 7.5. Protein concentration was determined by the Bradford method [24].

### 4.3. Iota Toxin Production and Purification

Ia and Ib components of iota toxin were produced and purified as previously described [23]. Briefly, *iap* and *ibp* from *C. perfringens* NCIB10748 were cloned into the *Escherichia coli-C. perfringens* shuttle vector pJIR750 yielding pMRP147 and pMRP384 which have been transfected by electroporation into the lecithinase-negative *C. perfringens* strain 667. The recombinant *C. perfringens* strains were grown and the culture supernatants were processed as indicated above. Ia and Ib were purified by DEAE-Sephacel chromatography and gel filtration on Superdex200 as previously described [23].

Rabbit antibodies against purified Ia and Ib were produced and checked as previously described [25,26].

### 4.4. Western Blotting

*C. perfringens* concentrated culture supernatants Ia and Ib were run on a 10% SDS-PAGE and transferred onto nitrocellulose. After blocking with 5% nonfat dry milk in phosphate-buffered saline, the membranes were washed with Tris-HCl 10 mM ( pH 7.5) containing 150 mM NaCl and 0.1% Tween20 (TTS). The membranes were incubated with either rabbit anti-Ia or anti-Ib antibodies (1:3000 concentration) (Jackson Immunoresearch 111-035-006) for 1 h at room temperature. After three washes in TTS, membranes were incubated with horseradish peroxidase goat anti-rabbit immunoglobulins (1:3000) for 1 h at room temperature, and then processed for chemiluminescence with Immobilon Western (Millipore, Guyancourt 78280, France).

### 4.5. ADP-Ribosylation

For ADP-ribosylation, the following buffer was used: Tris 50 mM (pH 7.5) containing 5 mM MgCl_2_, 10 mM dithiothreitol, 10 mM thymidine, protease inhibitors (leupeptine 0.1 mM, pepstatin 1 mM, PMSF 2 mM), and biotin-NAD (Trevigen 4670-500-01) 12 μM. In vitro ADP-ribosylation was performed in 20 μl of the above buffer containing either 1 μg muscular (Sigma A-3653) or cellular (Cytoskeleton APHL95) actin, and 1 μL of concentrated *C. perfringens* culture supernatant. After 1 h incubation at 37 °C, the samples were run in SDS-PAGE, transferred onto nitrocellulose, processed with peroxidase streptavidin conjugate (Roche 11-089-153-001) 1:3000 for 1 h, and processed for chemiluminescence as in Section 4.4.

### 4.6. Cytotoxicity Assay

Vero (African green monkey kidney) cells were grown in Dulbecco’s modified Eagle medium (DMEM) supplemented with 10% fetal calf serum at 37 °C and 5% CO_2_. Cells were grown as confluent monolayers in 96-well plates. Then, the medium was changed to DMEM containing 0.1% bovine serum albumin and the cells were incubated with serial dilutions of *C. perfringens* culture supernatants. Changes in cell morphology characterized by cell rounding were microscopically observed after 18 h incubation.

### 4.7. Genome Sequencing, Assembly, and Analysis

Genomic DNA from *C. perfringens* strains was extracted and purified as previously described [27]. Whole-genome shotgun sequencing was performed using an Illumina NextSeq 500 sequencer. Libraries were constructed using Nextera XT technology and sequenced using a 2 × 150 nucleotide paired-end strategy. All reads were preprocessed to remove or correct artefactual or low quality bases. Sequenced reads were assembled using SPAdes (v. 3.11.0, St. Petersburg State University, St. Petersburg, Russian Federation, 2017) [28], and resulting scaffold sequences were annotated using Prokka (v. 1.11, University of Melbourne, Melbourne, Australia, 2014) [29]. Specific toxin genes (i.e., *becAB*, *colA*, *cpb*, *cpb2*, *cpd*, *cpe*, *etx*, *iap*, *ibp*, *netBEFG*, *pfoA*, *plc*, *tpeL*) were searched against genome sequences using BioNumerics (v. 7.6, Applied Maths NV, www.applied-maths.com, Sint-Martens-Latem, Belgium, 2016). In order to identify plasmid sequences within each assembly, the scaffold sequences were reordered against the complete chromosome of *C. perfringens* ATCC 13124 using Contiguator (v. 2.7.4, University of Florence, Florence, Italy, 2014) [30,31], and each non-ordered sequences was used as query to perform a BLAST search against all publicly available *C. perfringens* plasmid sequences (selection criteria: 90% similarity and 70% query coverage).

### 4.8. Phylogenetic Reconstruction

All publicly available *C. perfringens* genome assemblies were gathered from the NCBI repository (www.ncbi.nlm.nih.gov/genome/genomes/158) in order to build a recombination-purged core-genome using Parsnp (v. 1.1.2, National Biodefense Analysis and Countermeasures Center, Frederick, MD, USA, 2014) [32]. The resulting 1,211,302 aligned core nucleotide characters were analyzed using IQ-TREE (v. 1.6.7.2, University of Vienna, Wien, Austria, 2018) [33] to infer a maximum likelihood phylogenetic tree on 115 taxa with the evolutionary model GTR+F+R10 (automatically selected by IQ-TREE from the data).

### 4.9. Sequence Accession

The assembled genome sequences of strains 508.17 and 515.17 were deposited in the European Nucleotide Archive and are available under accession numbers UWOV01000001–UWOV01000120 and UWOU01000001–UWOU01000116, respectively.

## Figures and Tables

**Figure 1 toxins-11-00138-f001:**
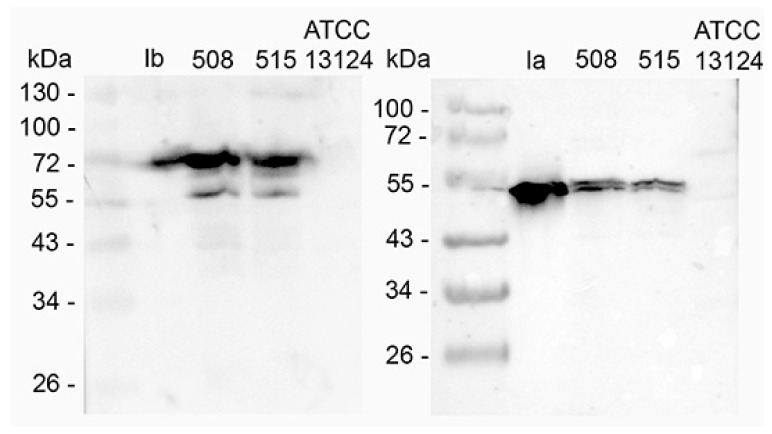
Production of iota toxin in *C. perfringens* 508.17 (508), 515.17 (515), and ATCC 13124.

**Figure 2 toxins-11-00138-f002:**
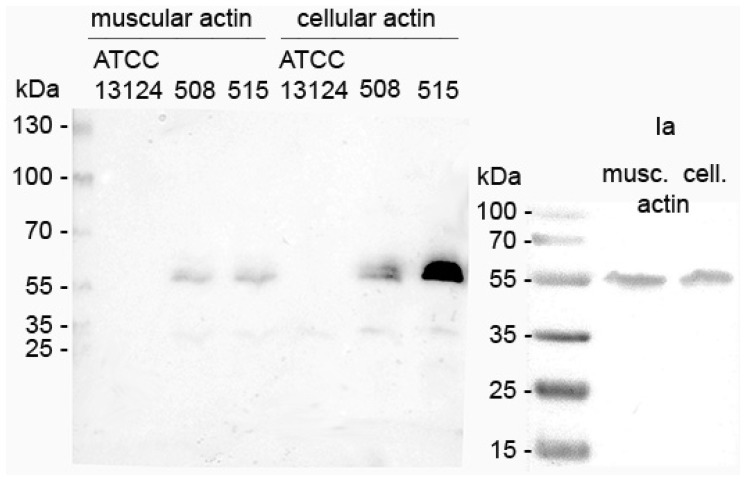
In vitro ADP-ribosylation of muscular and cellular actin with *C. perfringens* 508.17 (508) and 515.17 (515). ADP-ribosylation with purified Ia from NCIB10748 is shown as a positive control. No actin ADP-ribosylation was observed with the type A *C. perfringens* strain ATCC 13124.

**Figure 3 toxins-11-00138-f003:**
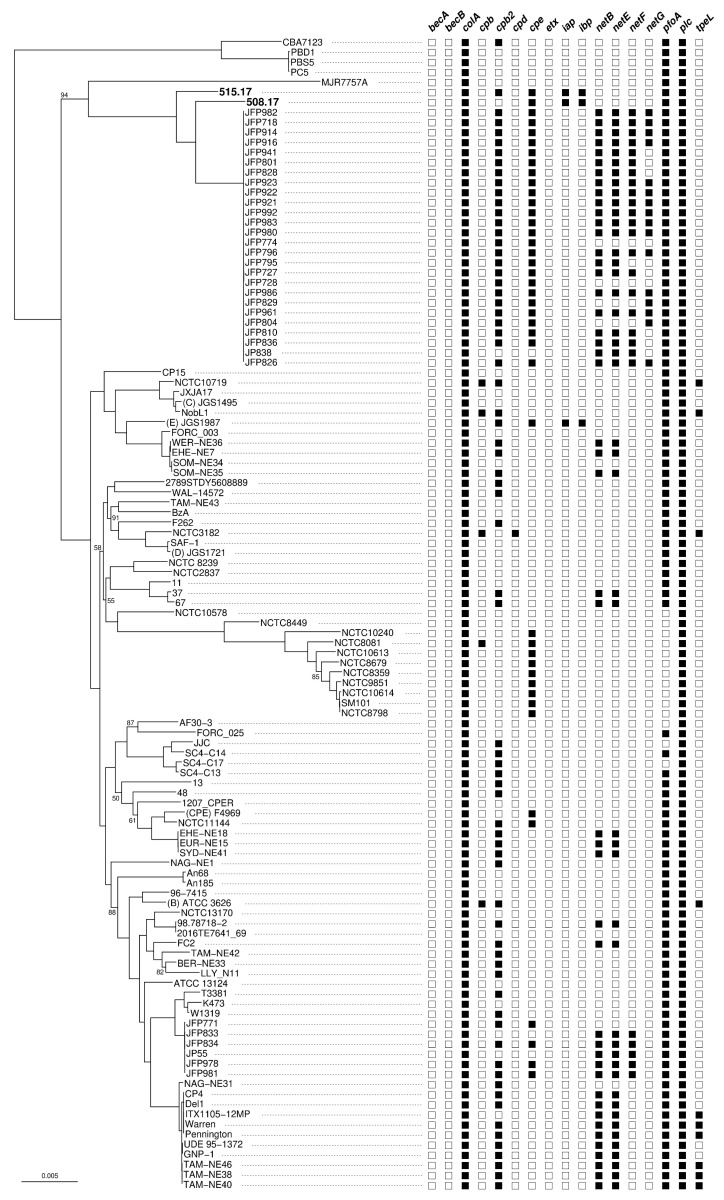
Maximum likelihood phylogenetic tree of *C. perfringens* based on their core genome and presence/absence pattern of 17 toxin genes. Each branch support was assessed by the ultrafast bootstrap approach (only support values <95% are shown). The scale bar represents 0.005 substitutions per nucleotide. The presence/absence of each of the 17 toxin genes is indicated by a black/white square for each of the 115 analyzed genomes. *becAB*: binary enterotoxin of *C. perfringens*; *colA*: collagenase A; *cpb*: *C. perfringens* beta toxin; *cpb2*, *cpd*: *C. perfringens* delta toxins; *cpe*: *C. perfringens* enterotoxin; *etx*: *C. perfringens* epsilon toxin; *iap*, *ibp*: *C. perfringens* iota toxin; *netBEFG*: necrotic enteritis toxins; *pfoA*: perfringolysin; *plc*: *C. perfringens* alpha toxin; *tpeL*: toxin perfringens large cytotoxin.

**Table 1 toxins-11-00138-t001:** Amino acid (aa) sequence comparison of iota toxin components Ia and Ib from 508.17 and 515.17 strains versus classical iota toxin (X73562) from NCIB10748 and iota variant from pCPPB-1.

Strain/Sequence	Iota Ia Similarity % (aa)				Iota Ib Similarity % (aa)			
	508.17	pCPPB-1	515.17	X73562	508.17	pCPPB-1	515.17	X73562
508.17	100.00				100.00			
pCPPB-1	99.87	100.00			99.85	100.00		
515.17	99.73	99.86	100.00		99.93	99.86	100.00	
X73562	91.20	91.33	91.19	100.00	89.69	89.77	89.77	100.00

**Table 2 toxins-11-00138-t002:** Comparative alignment and amino acid similarity of CPE 508.17 and 515.17 versus pCPPB-1 variant CPE as well as pCPF4969 and pJFP838-all classical plasmid CPE. The amino acid (aa) changes between variant and classical CPE are indicated. CPE sequences from 508.17 and 515.17 match with that of pCPPB-1 and diverge at 12 positions from classical CPE.

Strain/Sequence	Variable aa Positions on *cpe*												Similarity %			
	18	20	32	172	193	217	257	275	276	283	287	313	508.17	515.17	pCPPB-1	pCPF4969
508.17	L	V	K	A	S	T	R	Q	E	N	I	A	-			
515.17	L	V	K	A	S	T	R	Q	E	N	I	A	99.07	-		
pCPPB-1	L	V	K	A	S	T	R	Q	E	N	I	A	100.00	99.07	-	
pCPF4969	F	I	N	G	T	S	K	E	Q	K	V	S	96.36	97.3	96.36	-
pJFP838-all	F	I	N	G	T	S	K	E	Q	K	V	S	96.36	97.3	96.36	100.00

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
