# Peer review of "Two Clostridium perfringens Type E Isolates in France"

_toxins, 2019, doi:10.3390/toxins11030138_

Reviewer 1 Report

The authors have fully addressed my earlier criticisms and comments. 

Author Response

Thanks for your comments

Reviewer 2 Report

The species names are still not italicized in the Abstract, Keywords and Key Contribution sections.

It would be good to include a previously identified and characterized type E strain(e.g. NCIB10748) as a positive control in Figure 1.

The positive controls of Figure 2 should be run on the same gel as the test samples.

Ln64. What does "seems to contain at least six plasmids" mean? Where there six plasmids detected or possibly more? Where they visualized on gels? Where six plasmid contigs identified in the genome assembly? Where six distinct plasmid replication regions identified? Where any of the plasmid contigs closed to give circular plasmid sequences? Are the plasmid sequences included in the sequence information deposited with ENA? If there are six plasmids why list seven plasmids that they are similar to? Perhaps a table showing percentage plasmid homology to known plasmids would be useful?

Ln71. Give references for BEC and CPILE.

Ln153. Include centrifugation conditions used to collect precipitate.

Ln154. Should a reference be included for the Bradford method?

Ln218. Include volume and pages for reference.

Ln230. Italicize species name.

Lns215-252. Don't italicize internal words. (also Lns272-273, 276-277)

Author Response

Reviewer 2

The species names are still not italicized in the Abstract, Keywords and Key Contribution sections.

 The species names were italicized in our original document that we have upload. We have checked again that the species names are italicized.

It would be good to include a previously identified and characterized type E strain(e.g. NCIB10748) as a positive control in Figure 1.

The Ia and Ib proteins in Figure 1 as positive controls are from NCIB10748. They have been produced in C. perfringens with a multicopy plasmid containing the original iap and ibp genes under their own promoter.

The positive controls of Figure 2 should be run on the same gel as the test samples.

 Since it was difficult to calibrate the same revelation intensity of the ADP-ribosylation with purified Ia and C. perfringens crude supernatants, the enzymatic ADP-ribosylations that have been done in the same conditions with Ia and C. perfringens supernatants, have been load on two separate gels and were revealed separately to get readable and interpretable results.

Ln64. What does "seems to contain at least six plasmids" mean? Where there six plasmids detected or possibly more? Where they visualized on gels? Where six plasmid contigs identified in the genome assembly? Where six distinct plasmid replication regions identified? Where any of the plasmid contigs closed to give circular plasmid sequences? Are the plasmid sequences included in the sequence information deposited with ENA? If there are six plasmids why list seven plasmids that they are similar to? Perhaps a table showing percentage plasmid homology to known plasmids would be useful?

This sentence was very incorrectly written because we only focused on the first BLAST hit for each scaffold sequence that was not reordered against ATCC 13124 (see 4.7).

Identification of the putative plasmid scaffold sequences was more carefully performed and is now summarized with the corresponding sentence in the manuscript:

The two de novo assemblies are quite similar at first sight, i.e. 3.47 and 3.58 Mbs, 28.00% and 28.06% GC-content, 3,085 and 3,286 CDS for isolates 508.17 and 515.17, respectively. However, scaffolds from isolate 508.17 contain only one sequence that is almost identical to pCPPB-1 (accession number UWOV01000018; 99.8% similarity), whereas those from isolate 515.17 contain 19 sequences that could be reordered against pCPPB-1 (i.e. UWOU0100067, 108, 087, 023, 106, 043, 047, 005, 018, 003, 036, 090, 086, 068, 013, 006, 014, 007, 101; 99.7% identity), and 7 additional sequences showing strong BLAST similarity (e.g. > 99.7% identity) with pJFP838D and pD13118_cpe (i.e. UWOU0100008, 009, 016, 022, 024, 026, 034). This indicates that the strain 515.17 contains two additional plasmids related to pJFP838D and pD13118_cpe or a plasmid combining these sequences..”

Of note, only sequence similarity searches were performed, and corresponding GenBank accession numbers are provided.

Ln71. Give references for BEC and CPILE.

References have been added

Ln153. Include centrifugation conditions used to collect precipitate.

The centrifugation conditions have been added

Ln154. Should a reference be included for the Bradford method?

The reference of the Bradford method has been added

Ln218. Include volume and pages for reference.

The reference has been updated

Ln230. Italicize species name.

The correction has been done

Lns215-252. Don't italicize internal words. (also Lns272-273, 276-277)

The correction has been done (don't capitalize instead of italicize)

Reviewer 3 Report

there are no major comments

Author Response

Thanks for your comments